# The Comparison of the Effects of Nodular Cast Iron Laser Alloying with Selected Substances

**DOI:** 10.3390/ma15217561

**Published:** 2022-10-28

**Authors:** Marta Paczkowska

**Affiliations:** Department of Transport and Civil Engineering, Institute of Machines and Motor Vehicles, Poznan University of Technology, Piotrowo 3, 60-965 Poznan, Poland; marta.paczkowska@put.poznan.pl

**Keywords:** cast iron, surface layer, laser treatment, hardness, microstructure

## Abstract

The aim of this research was to compare the effects of laser treatment, with the same heating conditions, using four selected alloying substances (silicon, cobalt, silicon nitride and titanium), in the surface layer of nodular cast iron. The treatment was performed with a molecular laser. As the microstructure observation revealed, the greatest amount of implemented elements was diluted during the treatment in a solid solution. In all cases (except during the alloying process with cobalt), in the alloying zone, a fine and homogeneous microstructure was found. In the alloying zone, cobalt counteracted the formation of the martensitic microstructure so effectively that austenite turned into exclusively fine perlite (or bainite at most). The size of the obtained alloyed zone was different, despite the same laser heat treatment parameters. A 30% smaller depth of zone after laser alloying with silicon nitride, as compared with alloying with cobalt or silicon, was observed. The highest strengthening of the alloyed zone could be expected when silicon (hardness was approx. 980HV0.1 and the modulus of elasticity was 208 GPa) and titanium (hardness was approx. 880HV0.1 and the modulus of elasticity was 194 GPa) were used. The lowest hardness (700HV0.1) was observed for the zone alloyed with cobalt due to pearlite (or bainite) existence.

## 1. Introduction

Nodular cast iron has been widely used in fields such as the automotive industry, as well as a multitude of others. Good mechanical properties related to its lower density (compared with other iron alloys such as steels), as well as its vibration damping ability, render this material suitable for many engines as well as brake parts. Some of these parts (such as the crankshaft, camshaft, and brakes) are subjected to intense tribological wear, corrosion or thermal shocks; therefore, specific properties of their surface layers are required.

For example, a laser beam can be used as a heat source to change the microstructure of the near surface areas of such parts. Such a treatment is used in surface engineering to modify the properties of the surface layer. Modifying the surface layer by heating with a laser beam is a method of improving the surface properties of a material, and it is a method that has been advancing since the end of the last century [1]. Laser processing allows significant changes in the microstructure and properties of local surface areas, including cast iron [2,3,4,5,6,7,8,9]. It is noteworthy that in the case of laser processing, the conditions for microstructure formation are completely different compared with conventional hardening methods. The process is characterized by extremely fast cooling rates [7]. The microstructure of gray cast iron, formatted in the surface region as a result of a high melting rates and ultrafast re-solidification rates during laser heat treatment is often referred to as ledeburitic [8]. It was confirmed that gray cast irons, after surface hardening in this manner, are characterized by better wear resistance compared with untreated ones [9].

Hardness, wear resistance as well as corrosion resistance can be increased by applying an element or substance during laser processing and performing an alloying of the surface layer [10,11,12,13,14,15,16,17,18,19,20,21,22,23,24]. One of the elements that can increase hardness and wear resistance is silicon. Preliminary studies on laser surface alloying of mild steel with silicon have shown a significant improvement in its surface properties [16]. In the alloying zone, dispersed and hard silicides such as FeSi, Fe_2_Si and Fe_3_Si were found [17]. It is noteworthy that silicon also increases resistance to tempering. In addition, deoxidation was observed in the alloying zone during the laser treatment with a mixture of Si_3_N_4_, Fe, and Cr. It was found that the higher the Si content in the alloying mixture, the better the deoxidation [18]. Nitrogen is another alloying element that can be added to the surface layer to improve its properties. Research on the laser nitriding of steel parts has shown that, by this method, high hardness, good wear, and corrosion resistance can be achieved [19,20,21]. Research on steel has shown that depending on the parameters of the laser treatment, a two to four fold increase in the hardness of the alloy layer can be reached. N_2_ gas was used for the procedure [20]. Other studies (on low-carbon steel) [21] have shown that laser alloying with nitrogen is possible, using only this element from the atmosphere. It is also possible to use Si_3_N_4_ powder as a source of nitrogen. In the case of alloying with a mixture of Si3N4, Fe, and Cr, a decomposition of Si_3_N_4_ was found (in the paper mentioned above [18]). This type of laser treatment allowed for the achievement of an almost three fold increase in hardness. It is noteworthy that titanium is also used for laser alloying. The TiC phase can be formed in the surface layer of steel by laser alloying with titanium [22]. Given that there is more carbon in cast iron than in steel, an increased number of such hard phases, after laser alloying with titanium, could be formed using this cast material. Such a reaction between carbon and titanium was observed after the cast iron laser treatment and after plasma titanium spraying [23]. Titanium, as an alloying substance, has been also used in research on ductile iron [14]. An element such as cobalt can also be used in the case of laser surface modification. Studies [24] on the laser alloying of steel with cobalt have shown that, in this sense, it is possible to achieve high hardness at elevated temperatures. The tests have shown that the thermal fatigue strength increased by 160% (compared with the base material). The average cobalt content in the alloying zone was 3 to 6%.

Thus, elements such as silicon, nitrogen, and titanium, as well as cobalt, seem to be reasonable additives in laser alloying treatments, and they have the ability to increase certain properties of the surface layer such as hardness, wear, corrosion or even the heat resistance of machine parts that are treated using this process. Laser alloying treatments that use those elements are the subject of many studies, and they explain the influence of particular additives on the surface layer; however, the results are always assessed under laser treatment certain conditions. Hence, it is difficult to evaluate the differences between the effects of particular additives during laser alloying treatments in the surface layer; therefore, the aim of this exploratory research was to determine the differences between the effects on the surface layer after the laser treatment, with the same heating conditions, but with four different, selected substances (silicon, cobalt, silicon nitride, and titanium). Such a comparison should allow one to determine the advantages and disadvantages of these substances when they are used as alloying materials for the laser surface treatment of nodular cast iron.

## 2. Materials and Methods

Nodular iron 600-3 was selected as the investigation material. The chemical composition of the tested iron was as follows: C = 3.0% by weight, Si = 2.87% by weight, Mn = 0.32% by weight, P = 0.035% by weight, S = 0.013% by weight, Cr = 0.003 wt.% Cu = 0.559 wt.%, Al = 0.009 wt.%, Mg = 0.38 wt.%. Its hardness was 217 HBr.

The dimensions of the sample for the laser treatment were as follows: 25 mm × 15 mm × 5 mm. The samples were covered with an alloying paste on one side. The covering paste consisted of an alloying substance and a water glass. Details of the alloying substances are shown in Table 1. The coated nodular iron samples are shown in Figure 1.

Additionally, in order to compare the effects of laser alloying, one sample was remelted with the same laser beam parameters (but without any alloying additives). In the case where remelting occurred without alloying, the covering paste consisted of black gouache.

The treatment was carried out using a molecular CO_2_ continuous Trumph laser (type TLF 2600 t) with the TEM01 mode. With each variant, a constant interaction time of the laser beam (t = 3.47 s), and the power of the laser beam (P = 1 kW), with the fluence of the laser beam (F = 102 J/mm^2^), were applied.

The temperature on the surface of the samples was controlled during the laser treatment, with two pyrometers (Raytek Marathon Series) that had measuring ranges between 300 and 2000 °C (1) and 700 and 3000 °C (2), respectively; this is because the temperature can exceed 2000 °C.

The possibility of exceeding 2000 °C during the treatment with the applied beam parameters has been shown using certain calculations. The temperature, as well as the cooling rate distributions from the surface to the core material, were calculated using the equations presented in Ref. [1] and shown in Figure 2. The temperature distributions from the surface to the core material during the laser treatment were considered a function of the laser treatment parameters (laser beam: power, radius, interaction time) and the physical properties of the treated material (density, specific heat, thermal conductivity, thermal diffusivity); these were evaluated similarly to the nodular iron laser boronizing case [11].

The treatment samples were cut perpendicularly to the alloyed zone, as shown in Figure 3. The results of the laser processing were analyzed using a scanning electron microscope and an optical microscope (zone geometry dimension assessment and microstructure examination, samples were etched with nital 3%), a Vickers hardness tester (the indentation load was 0.9807 N and 5 repetitions of the test were conducted per sample), an X-ray diffractometer for phase identification (a monochromatic radiation of a lamp with a copper anode (λ = 1.54 A) was applied), an AES (nitrogen, titanium, cobalt identification) and an EDS (silicon identification). An NHT nanoindentation tester with a Berkovich diamond indenter was also used (enabling an analysis of selected nanomechanical properties such as the elasticity modulus, the creep modulus, and an estimation of the value of the plastic and elastic work in the modified layer). The maximum load was 100.00 mN for 5 s. 5 repetitions were performed.

## 3. Results and Discussion

The result of the surface temperature measurement using two pyrometers is shown in Figure 4. It can be observed that the time of heating did not exceed 4 s, which renders the laser beam interaction time as 3.47 s. It can also be observed (based on the high range pyrometer) that the heating can be divided into stages. This is due to the TEM01 mode of the laser device (the distribution of the laser beam power on the section is characterized by two maximums of its value).

Remelting the surface layer started in the first stage when the average temperature very quickly reached approx. 2500 °C. This value confirms the temperature calculations (Figure 2a). The temperature in the second stage reached approx. 1700 °C and the surface was chilled.

The average cooling rate of the surface obtained from the pyrometers was approx. 500 °C/s, which makes it similar to the one calculated from the equation (Figure 2b). This section may be divided into subcategories. It should provide a concise and precise description of the experimental results, their interpretation, as well as the experimental conclusions that can be drawn.

After the heat treatment with the laser, it was possible to distinguish the alloyed zone of the nodular iron in all cases. An example of a cross-section of the surface layer, with a range of modifications, is shown in Figure 5.

The transition zone and the hardened (from the solid state) zone were formed under the alloyed zone. This is a typical microstructure after laser alloying or melting this type of cast iron [2,3]. The first zone from the surface is generally more homogenous and finer, compared with others and the bulk material.

Of crucial significance for the surface layer is the alloyed zone. This zone is usually characterized by the highest hardness and is responsible for the wear and corrosion resistance; therefore, its size (depth and width) is significant when planning and performing the laser heat treatment. The measurement of the alloyed zone dimensions revealed that they were different for different alloying substances (Figure 6, Figure 7, Figure 8 and Figure 9). Figure 10 presents the width and depth of the alloyed zone (for comparison, only the width and depth of the zone that remelted, without alloying, was presented). Four samples per each variant have been tested. It appeared that for the alloying with cobalt, silicon, or titanium the width (~4.5–5 mm) is comparable to the width of the zone, but only after laser remelting (covered before laser melting with gouache). The depth of the zones after alloying with cobalt and silicon is comparable to the depth of the zone, but only after remelting (~0.85->0.9 mm). Despite the same laser treatment conditions, an evident difference was visible for the zone after laser alloying with silicon nitride (the width was <4 mm and the depth <0.7 mm). The smaller alloyed zone in this case is a result of the color of the alloy substance (Figure 1). A 30% smaller depth of the zone after laser alloying with silicon nitride, compared with alloying with cobalt or silicon, is a result of the different absorptions of laser beam radiation.

The presence of the implemented elements was confirmed by the AES and EDS methods. In the alloyed zone, all alloying elements were detected. The auger electron spectroscopy revealed that there was approx. 2%at. of nitrogen in the zone alloyed with silicon nitride, 2.5%at. of titanium in the alloyed zone after laser alloying with titanium, and 6%at. of cobalt in the alloyed zone after laser alloying with cobalt. Due to similar peaks in the Auger electron energy spectrum for iron and silicon, Si identification is difficult in iron alloys; hence, the EDS method was applied for this element. Approx. 11%at. Si was detected (Figure 11).

The microscopic observations indicate that the implemented alloying elements were mainly diluted in solid solutions. After laser alloying with silicon, a fine and dendritic microstructure was found. As a consequence of austenite crystallization that occurred directly from the liquid alloy, a microstructure similar to white hypoeutectic cast iron was formed (Figure 12).

After laser alloying with cobalt, the microstructure was coarser compared with the other alloyed zones (Figure 13). The microscopic observations did not reveal the martensite microstructure. This may be due to the reduced occurrence of supercooled austenite influenced by cobalt (this element remained in the austenite during the treatment, and it advanced the nucleation and growth transformation curves on TTT diagram). As a result, the austenite only turned into fine perlite (or bainite at most). No newly formed phases were recorded during the microstructure observations. This suggests the presence of cobalt, primarily in the solid solution.

After laser alloying with fine Si_3_N_4_, a dendritic microstructure was found. In this variant, as in the case of laser alloying with silicon, a microstructure similar to hardened white cast irons was formed (Figure 14).

A similar microstructure also appeared after laser alloying with titanium (Figure 15). Nevertheless, near the surface, after laser alloying with titanium, TiN or TiCN was observed (Figure 16). This is a result of titanium’s high affinity with nitrogen. During the laser treatment, nitrogen from the atmosphere was absorbed into the surface layer. TiCN was also observed in the experiment when titanium was used as an alloying substance during the laser treatment of ductile cast iron [14]. Titanium nitrides are often formed in austenitic, corrosive, resistible steels during welding (containing titanium).

The X-ray-based research revealed Feα and Fe_3_C in all the zones that were being analyzed (Figure 17, Table 2). Feγ was not found. The X-ray diffraction indicated a possibility of the presence of some newly formed phases containing the implemented elements. In the X-ray figures, some peaks matched the diffraction lines that are characteristic of individual phases (however, it should also be noted that some peaks do make superpositions).

In the case of laser alloying with silicon peaks that were derived from Fe_9_SiC_0.4_ and iron, silicides such as FeSi and FeSi_2_ were observed (the microscopic observations indicate that the implemented silicon should mainly be diluted in solid solutions). Phases such as Fe_2_Si or Fe_3_Si, that had previously been identified in the research on steel laser alloying with silicon, were not detected [15].

In the alloyed zone containing cobalt, some peaks of Co_3_C cobalt carbide were observed. In the case of this element, it is noteworthy that the mutual substitution in the lattice of iron and cobalt can take place due to the comparable size of their atoms. Moreover, some peaks of oxides, such as CoFe_2_O_4_, Fe_2_O_3_, and Fe_3_O_4_, were also visible.

On the other hand, peaks of FeN and FeN_0.03_ iron nitrides, and phases containing silicon such as FeSi_2_, FeSi and Fe_2_Si, were observed in the alloyed zone after the laser treatment with silicon nitride. The presence of these phases may suggest that the Si_3_N_4_ compound has (at least partially) decomposed. The high temperature in the surface layer, caused by the laser treatment, should allow the silicon nitride to decompose. Such a process was observed when steel was melted with a laser using this substance [18].

Peaks of oxides such as TiO_2_, (Fe_0.7_Ti_0.3_)O_5_, and Fe(TiO_3_), were revealed in the alloyed zone after laser alloying with titanium.

The performed laser treatment created a fine and more homogeneous microstructure in the alloyed zone (compared to the base material); it increased the hardness in all the investigated variants (Figure 18) and the hardness at least tripled.

Nevertheless, the hardness measurements showed some differences between variants. Laser alloying with silicon increased the average hardness by over 200HV0.1 in comparison to the hardness after laser remelting (without the alloying substance). In the case of laser alloying with silicon nitride or titanium, the increase in hardness was comparable (over 100HV0.1). Titanium likely did not increase the hardness in the same manner as silicon due to the fact that by combining with nitrogen, it formed areas (relatively small, but rich with this element and very hard—1500HV0.1) of TiN or TiCN, which were present in the vicinity of the surface. As a result, the amount in the rest of the alloying zone was relatively small. On the other hand, a smaller increase in hardness using silicon nitride (rather than silicon) may be the result of the insufficient decomposition of this substance in order to saturate the alloyed zone with silicon and nitrogen. The total decomposition of silicon nitride could be difficult even under laser treatment conditions. Due to the presence of pearlite in the microstructure of the zone alloyed with cobalt, the average hardness of this zone did not exceed the hardness of the zone obtained by laser remelting.

The nanoindentation measurements (Table 3) provided similar HV_IT_ hardness values (HV0.1) (Figure 18). The highest values of hardness were obtained for the zone that alloyed with silicon (except the area of TiN or TiCN in the case of alloying with titanium, where an even value of almost 1900HV_IT_ was detected). The maximum depth during the measurement, h_max_, was comparable to the zones that were alloyed with silicon and titanium (0.51 µm). A greater value was obtained for the cobalt (0.59 µm) and silicon nitride alloys (0.56 µm). For the area wherein TiN or TiCN was used, in the zone after alloying with titanium, the value obtained was the lowest (0.31 µm). In the case of the zone alloyed with silicon, the modulus of elasticity (E_IT_) and the elastic modulus (Er) obtained the highest values (the exception is the area wherein TiN or TiNC, in the zone alloyed with titanium, nearly reached 300 GPa). The highest value for plastic work (W_plast_) was obtained in the case of the zone alloyed with cobalt (7.72 nJ). The lowest value was for the zone alloyed with titanium (6.16 nJ). The value of elastic work (W_elast_) was comparable for all the cases. The value was slightly higher was for the zone alloyed with titanium. The creep parameter (C_IT_) was the highest for the zone alloyed with silicon (1.75%) and titanium (1.68%), whereas in the TiN or TiCN areas, it was characterized by the lowest value, which was 0.56%.

## 4. Conclusions

The following observations and comments can be made following the research carried out on the laser alloying of nodular iron with silicon, cobalt, silicon nitride, and titanium, using the same laser heat treatment parameters.

In the alloying zones, a fine and homogeneous martensitic microstructure (in case of alloying with silicon, silicon nitride and titanium), and fine perlite or at most binate (in case of alloying with cobalt), was found. Additionally, the X-ray diffraction showed the likely existence of Co_3_C, and some oxides, in the zone that alloyed with this cobalt. In the case of alloying with silicon, some peaks of Fe_9_SiC_0.4_, FeSi, and FeSi_2_ were found, and in the case of alloying with silicon nitride, some iron nitrides, such as FeN and FeN_0.03_, and phases containing silicon, such as FeSi_2_, FeSi, and Fe_2_Si, were revealed. Aside from areas of TiN or TiCN, in the zone alloyed with titanium, some oxides were also found. The attained thicknesses were lowest—less than 0.7 mm—in cases involving alloying with silicon nitride, and they were greatest—greater than 0.9 mm—in the zones that were alloyed with cobalt. A lower zone depth after laser alloying with silicon nitride, compared with the rest of the zones, was the result of different absorption levels regarding laser beam radiation.

The hardness of the obtained zones was over 800HV0.1 when alloyed with silicon nitride and titanium, and over 900HV0.1 for zones alloyed with silicon. The lowest hardness (700HV0.1) was observed for the zone alloyed with cobalt. For all the alloyed zones, HV_IT_ had a similar hardness value to the Vickers hardness value of HV0.1.

The modulus of elasticity (E_IT_) and the elastic modulus (Er) was highest in the case of the zone with silicon (208 GPa and 190 GPa, respectively). Those values were approx. 13% lower in the zones with cobalt or silicon nitride. The plastic work (W_plast_) was the highest in the case of the zone with cobalt (7.72 nJ). The lowest value occurred in the zone with titanium (6.16 nJ). The creep parameter (C_IT_) was the greatest for the zone alloyed with silicon and titanium (1.75% and1.68%, respectively).

The alloyed zone with the best properties was most likely to be the zone alloyed with silicon and titanium; this is due to the fact that it had the highest strength and a reasonably large modified zone.

## Figures and Tables

**Figure 1 materials-15-07561-f001:**
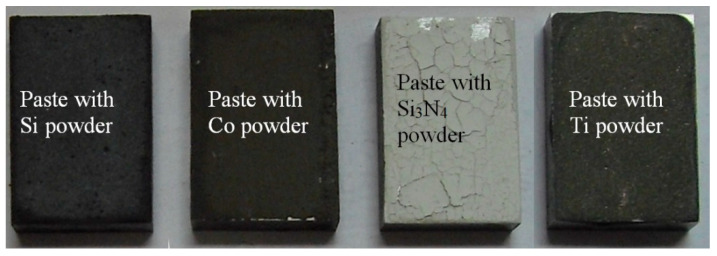
Samples of nodular iron covered with individual substances.

**Figure 2 materials-15-07561-f002:**
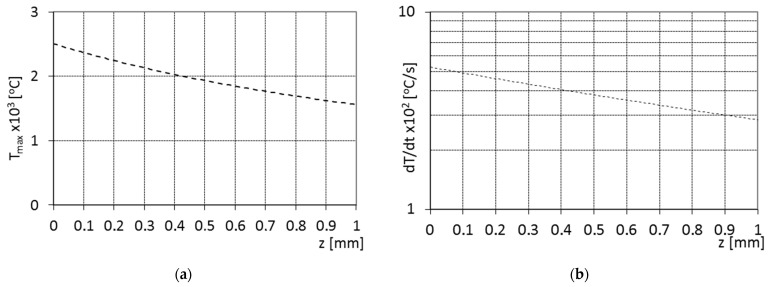
The calculated temperature (**a**) and the cooling rate (**b**) distribution in the distance from the surface to the core material (z) during the laser treatment with the applied laser beam parameters.

**Figure 3 materials-15-07561-f003:**
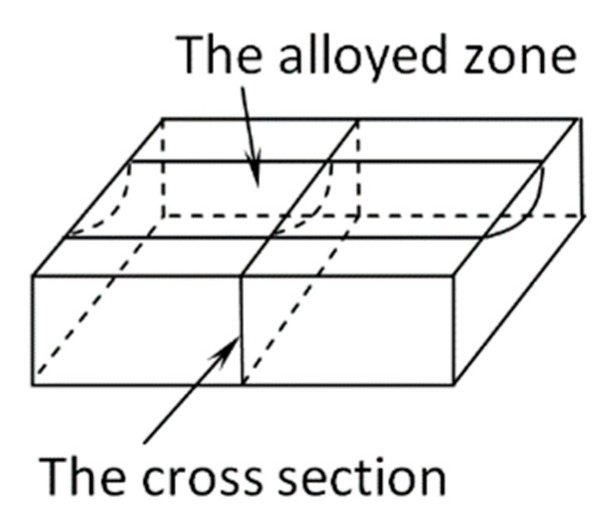
Scheme of the sample after the laser alloying with the cross-sectional plane marked for testing.

**Figure 4 materials-15-07561-f004:**
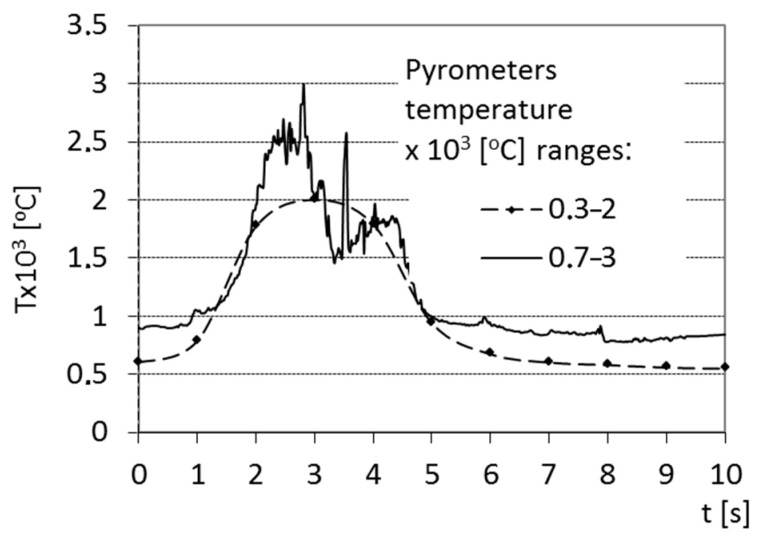
Temperature on the surface during the laser treatment recorded by two pyrometers.

**Figure 5 materials-15-07561-f005:**
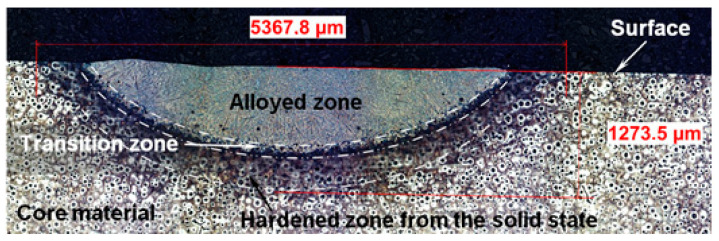
The cross-section of the surface layer of the nodular cast iron after the laser treatment (optical microscope, etched with nital 3%).

**Figure 6 materials-15-07561-f006:**
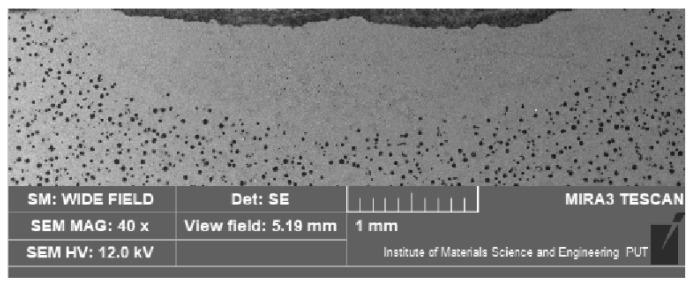
The cross-section of the surface layer of nodular iron after laser alloying with silicon (electron scanning microscope, etched with nital 3%).

**Figure 7 materials-15-07561-f007:**
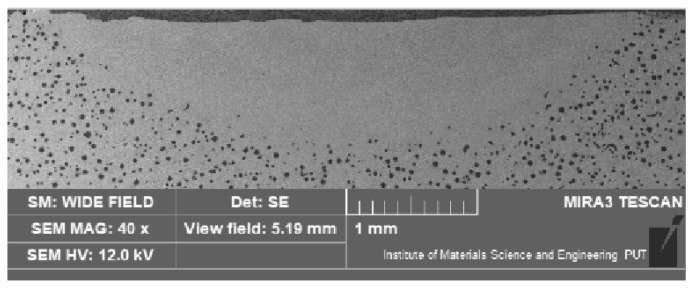
The cross-section of the surface layer of nodular iron after laser alloying with cobalt (electron scanning microscope, etched with nital 3%).

**Figure 8 materials-15-07561-f008:**
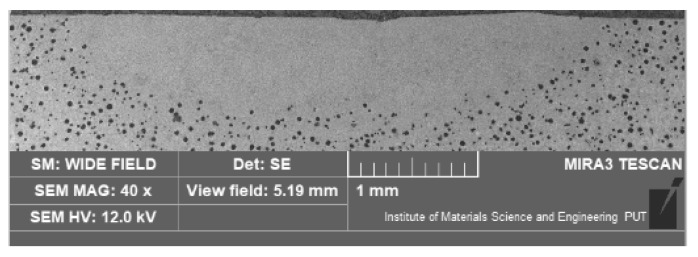
The cross-section of the surface layer of nodular iron after laser alloying with silicon nitride (electron scanning microscope, etched with nital 3%).

**Figure 9 materials-15-07561-f009:**
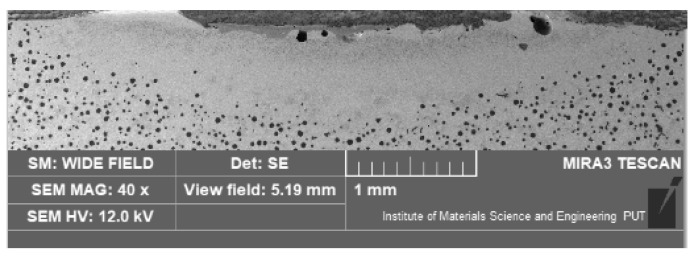
The cross-section of the surface layer of nodular iron after laser alloying with titanium (electron scanning microscope, etched with nital 3%).

**Figure 10 materials-15-07561-f010:**
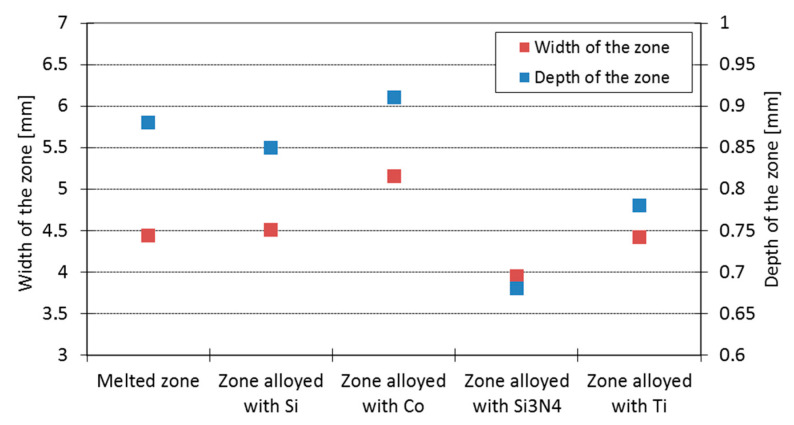
Dimensions of the melted zone and the alloyed zones after laser alloying with silicon, cobalt, silicon nitride, and titanium using the same laser beam parameters.

**Figure 11 materials-15-07561-f011:**
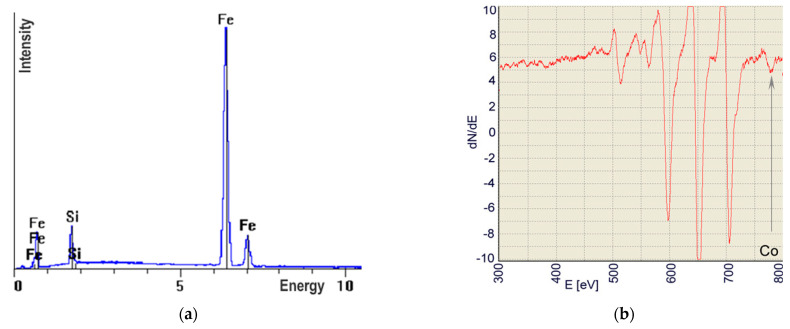
Examples of the AES and EDS spectra of the zones alloyed with: Si (**a**), Co (**b**), Si_3_N_4_ (**c**) and Ti (**d**).

**Figure 12 materials-15-07561-f012:**
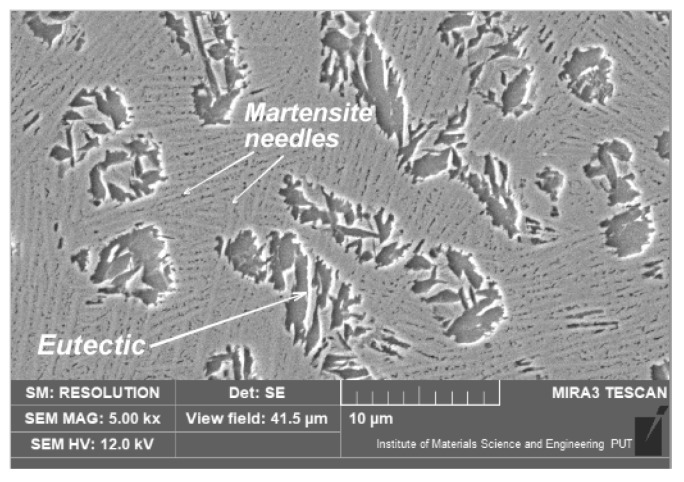
Microstructure of nodular cast iron surface layer, after laser alloying with silicon (scanning electron microscope, etched with nital 3%).

**Figure 13 materials-15-07561-f013:**
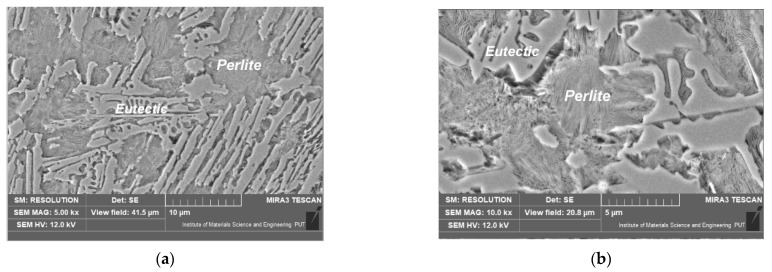
Microstructure of nodular cast iron surface layer, after laser alloying with cobalt (scanning electron microscope, etched with nital 3%). Lens magnification: (**a**) 5000×; (**b**) 10,000×.

**Figure 14 materials-15-07561-f014:**
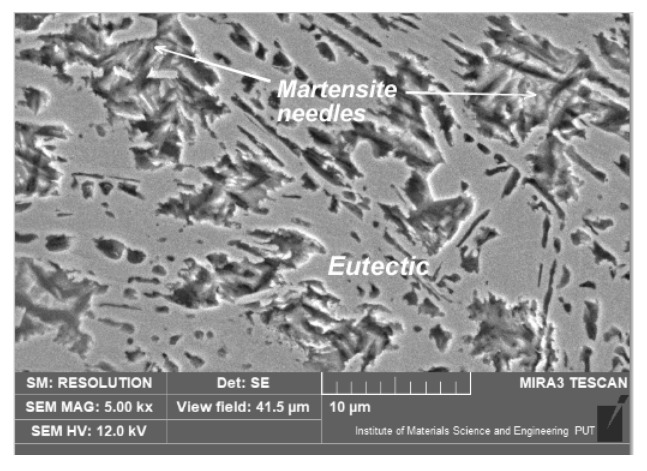
Microstructure of nodular cast iron surface layer, after laser alloying with silicon nitride (scanning electron microscope, etched with nital 3%).

**Figure 15 materials-15-07561-f015:**
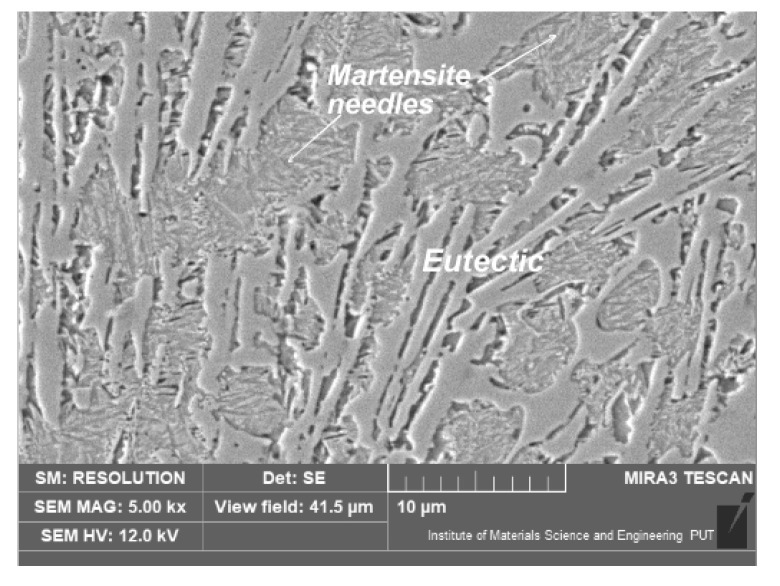
Microstructure of nodular cast iron surface layer, after laser alloying with titanium (etched with nital 3%).

**Figure 16 materials-15-07561-f016:**
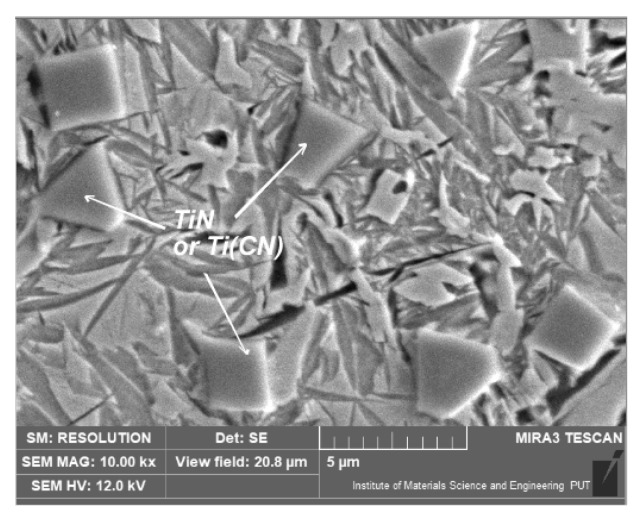
Microstructure of nodular cast iron surface layer, after laser alloying with titanium (scanning electron microscope, etched with nital 3%).

**Figure 17 materials-15-07561-f017:**
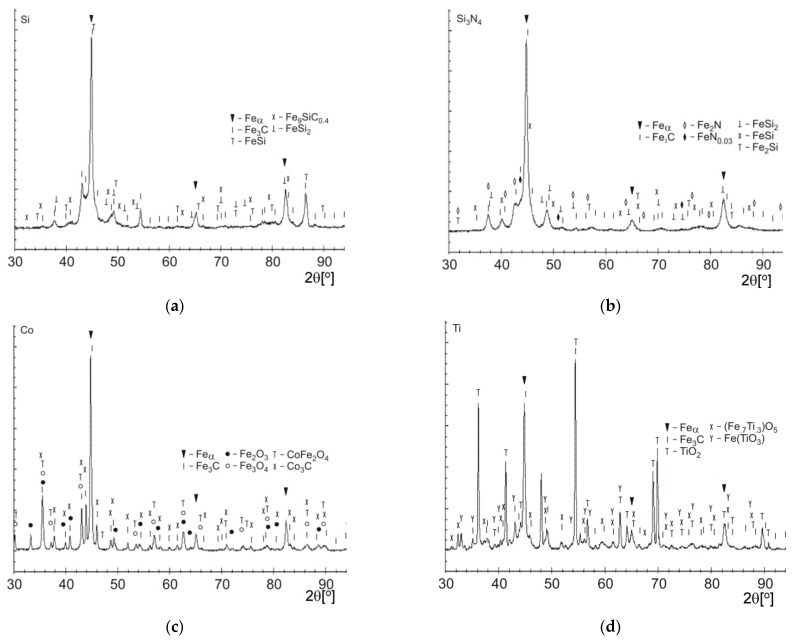
X-ray diffraction of nodular cast iron surface layer, after laser alloying with: (**a**) silicone; (**b**) silicon nitride; (**c**) cobalt; and (**d**) titanium.

**Figure 18 materials-15-07561-f018:**
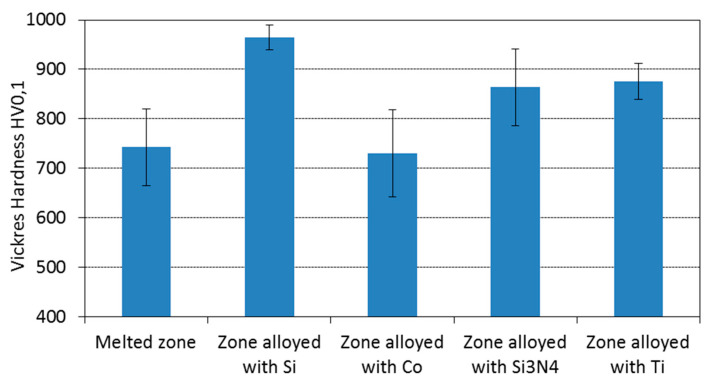
Average hardness of the melted and alloyed zones after laser alloying with silicon, cobalt, silicon nitride, and titanium.

**Table 1 materials-15-07561-t001:** The alloying substance particle size and its purity.

Alloying Substance	Particle Size	Purity
Si	325 mesh	99%
Co	2 μm	99.8%
α-Si_3_N_4_	325 mesh	≥99.9%
Ti	325 mesh	99%

**Table 2 materials-15-07561-t002:** X-ray diffraction results.

Model	Zone Alloyed with:			
	Si	Co	Si_3_N_4_	Ti
Feα	+	+	+	+
Fe_3_C	+	+	+	+
Fe_2_O_3_		+		
Fe_3_O_4_		+		
Fe_9_SiC_0.4_	+			
86–792 FeSi	+			
86–795 FeSi			+	
Fe_2_Si			+	
FeSi_2_	+		+	
TiO_2_				+
(Fe_0.7_Ti_0.3_)O_5_				+
Fe(TiO_3_)				+
CoFe_2_O_4_		+		
Co_3_C		+		
Fe_2_N			+	
FeN_0.03_			+	

**Table 3 materials-15-07561-t003:** Nanoindentaion test results.

Parameter	Zone Alloyed with:			
	Si	Co	Si_3_N_4_	Ti (TiN Area)
HV_IT_	978	696	783	965 (1873)
H_IT_ [GPa]	10.559	7.5135	8.4552	10.417 (20.22)
C_IT_ [%]	1.75	1.06	1.25	1.68 (0.56)
η_IT_ [%]	29.90	28.46	30.61	34.88 (44.19)
h_max_ [μm]	0.51	0.59	0.56	0.51 (0.39)
W_elast_ [nJ]	2.93	3.07	3.08	3.30 (2.93)
W_plast_ [nJ]	6.88	7.72	6.99	6.16 (3.70)
W_total_ [nJ]	9.81	10.79	10.08	9.46 (6.62)
Er [GPa]	190.86	165	168.36	179.9 (251.1)
E_IT_ [GPa]	208.37	175.39	179.58	194.17 (292.57)
E* [GPa]	228.98	192.73	197.34	213.37 (321.5)

## Data Availability

The data presented in this study are available on request from the author.

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
