# Peer review of "The Comparison of the Effects of Nodular Cast Iron Laser Alloying with Selected Substances"

_materials, 2022, doi:10.3390/ma15217561_

Round 1

Reviewer 1 Report

This paper compares the effects of surface alloying by laser remelting on nodular cast iron with different alloying elements. The work is interesting but to be published it is necessary to correct some lack of relevant information, and I think that the explanations in some of the results and important details of the microstructure could be improved, as well as the image quality of the microstructures. In addition, the author should specify the most relevant conclusions of the work.

Materials and Methods

1.    In this section you must indicate the load applied to obtain the Vickers       hardness, regardless of whether you indicate it in the results section. At the   same time, you must also indicate the number of tests per sample.

2.    When you analyse the microstructure you must put the attack reagent used (see line 115).

3.    In several studies, we have found that the Cu tube has difficulty in capturing the peaks emitted at low intensity. For this reason, it is sometimes advisable to use a Mo tube to avoid fluorescence, reducing the background intensity so that the reflection is not camouflaged. Have you taken this into account? It is advisable to indicate the X-ray diffractometer emitting tube (Cu, Mo or other) in the manuscript when you mention the X-ray diffractometer test in this section (see line 116).

Results and discussion

1.    Before figure 5, I would greatly improve the manuscript by adding a figure showing the images of the different layers with their width and depth, as there are considerable differences.

2.   In lines 181 and 183, you state that no martensite is observed. Perhaps this could be due to the fact that Co remains in solid solution in the austenite and advances the nucleation and growth transformation curves, reducing the hardenability of the austenite.

3.    The clarity of the microstructural section needs to be improved.

The quality should be improved, images should be added at higher magnifications and phases should be marked with arrows (to show the existence of martensite and pearlite/bainite) and precipitates in the case of figure 11.

My suggestion would be:

The figures corresponding to the microstructures of the four substrates are logically dendritic, being a structure with conditions similar to the cast state. For this reason, at low magnification, they do not provide much important information.

However, the microstructure at higher magnifications would demonstrate the important facts explained in the results. In figure 7, it is convenient to provide an image at higher magnification by marking with arrows the different phases observed (carbide and martensite). The same with figures 9 and 10. Figure 8 (b) needs a higher resolution and image quality to clearly see the differences with the rest (pearlite/bainite phases) and to mark these phases with arrows. Figure 11 needs to improve the quality and to point out with arrows the Ti nitrides and carbonitrides.

4.    In lines 242 to 245, it is mentioned that the hardness after alloying the melt with Si, increased 200HV, whereas if this happens with silicon nitride or Ti it increases 100HV. Is the possible explanation for this fact known? If so, please discuss it.

Conclusions

1.    The wording is excessively long. You include comments and explanations that are part of the results. You should simplify it a lot and focus on the most relevant conclusions you get from the work.

Author Response

Dear Reviewer,

Thank you for your valuable comments and suggestions. I hope, I have explained everything and corrected in accordance with your recommendations.

I have put the response under your comments and questions.

Comments and Suggestions for Authors

This paper compares the effects of surface alloying by laser remelting on nodular cast iron with different alloying elements. The work is interesting but to be published it is necessary to correct some lack of relevant information, and I think that the explanations in some of the results and important details of the microstructure could be improved, as well as the image quality of the microstructures. In addition, the author should specify the most relevant conclusions of the work.

Materials and Methods

  1. In this section you must indicate the load applied to obtain the Vickers       hardness, regardless of whether you indicate it in the results section. At the   same time, you must also indicate the number of tests per sample.

Done.

The indentation load was 0.9807N and 5 repetitions of test were made per sample.

  1. When you analyse the microstructure you must put the attack reagent used (see line 115).

I have added information about the reagent (it was a nitric acid solution in ethanol) to all figures presenting the microstructures. Figure 1 presents just photography of samples covered with different pastes.

  1. In several studies, we have found that the Cu tube has difficulty in capturing the peaks emitted at low intensity. For this reason, it is sometimes advisable to use a Mo tube to avoid fluorescence, reducing the background intensity so that the reflection is not camouflaged. Have you taken this into account? It is advisable to indicate the X-ray diffractometer emitting tube (Cu, Mo or other) in the manuscript when you mention the X-ray diffractometer test in this section (see line 116).

I have missed this information during writing. Thank you. Now, it is added. Unfortunately, I had to use a copper anode because I had access only to the diffractometer with such test parameters.

Results and discussion

  1. Before figure 5, I would greatly improve the manuscript by adding a figure showing the images of the different layers with their width and depth, as there are considerable differences.

I have added figures (now in the manuscript the numbering of the figures has changed).

  1.  In lines 181 and 183, you state that no martensite is observed. Perhaps this could be due to the fact that Co remains in solid solution in the austenite and advances the nucleation and growth transformation curves, reducing the hardenability of the austenite.

Yes, you are right. I tried to explain it by following statement: “This may be due to the reduction of the range of occurrence of supercooled austenite influenced by cobalt.” But, now I have corrected and try to clarify it more clearly like you specified in your sentence.

  1. The clarity of the microstructural section needs to be improved.

The quality should be improved, images should be added at higher magnifications and phases should be marked with arrows (to show the existence of martensite and pearlite/bainite) and precipitates in the case of figure 11.

My suggestion would be:

The figures corresponding to the microstructures of the four substrates are logically dendritic, being a structure with conditions similar to the cast state. For this reason, at low magnification, they do not provide much important information.

However, the microstructure at higher magnifications would demonstrate the important facts explained in the results. In figure 7, it is convenient to provide an image at higher magnification by marking with arrows the different phases observed (carbide and martensite). The same with figures 9 and 10. Figure 8 (b) needs a higher resolution and image quality to clearly see the differences with the rest (pearlite/bainite phases) and to mark these phases with arrows. Figure 11 needs to improve the quality and to point out with arrows the Ti nitrides and carbonitrides.

Yes, You are right. I have made new figures and marked phases.

  1. In lines 242 to 245, it is mentioned that the hardness after alloying the melt with Si, increased 200HV, whereas if this happens with silicon nitride or Ti it increases 100HV. Is the possible explanation for this fact known? If so, please discuss it.

This is a very interesting observation. It is hard to explain it clearly. Alloying additives increase hardness to a different extend and in different ways, additionally in such exceptional conditions occurring in case of laser treatment. Perhaps, Ti did not increase the hardness as much as Si because it formed areas (relatively small but rich with this element) of TiN or TiCN presence in the vicinity of the surface by combining with nitrogen. So its amount in the rest of the alloying zone was relatively small (not so much as to increase the hardness as in the case of Si). On the other hand, in the case of smaller increase of hardness using silicon nitride (than Si), it could be due to, that this substance perhaps did not decompose enough to saturate the alloying zone with silicon and nitrogen. The total decomposition of silicon nitride could be difficult even under laser treatment conditions.

I added some discussion also in the manuscript.

Conclusions

  1. The wording is excessively long. You include comments and explanations that are part of the results. You should simplify it a lot and focus on the most relevant conclusions you get from the work.

Corrected

You are absolutely right. I have corrected.

Reviewer 2 Report

This paper focus on the comparison of the effects of nodular cast iron laser alloying with selected substances. The overall feeling is that the paper is not innovative enough, and it is more of a description of experimental results, lacking depth. The following comments should be considered.

1. The introduction does not clearly explain the main problems of the current research, and does not propose the research purpose of this paper based on the literature analysis.

2. page3: Figure 3 should be Figure 2. How to ensure the reliability of the calculated temperature?

3. The experimental details are not described in detail, such as sampling location, sample preparation method, etc.

4. Page4: “the heating can be divided into stages. This is due to the TEM01 mode of the laser device.” Please explain TEM01.

5. Figure 5: How many samples of each condition were observed to guarantee the persuasiveness and reliability of the dimensions of the melted zone and the alloyed zones?

6. What are the abscissa and ordinate of Figure 6a?

7. The image quality is too low, such as Fig8b and Fig 11.

8. The conclusion is poorly written. The content is too long and repeats the previous results. The authors are advised to summarize the results.

Author Response

Dear Reviewer,

Thank you for your valuable comments and suggestions. I hope, I have explained everything and corrected in accordance with your recommendations.

I have put the response under your comments and questions.

Comments and Suggestions for Authors

This paper focus on the comparison of the effects of nodular cast iron laser alloying with selected substances. The overall feeling is that the paper is not innovative enough, and it is more of a description of experimental results, lacking depth. The following comments should be considered.

  1. The introduction does not clearly explain the main problems of the current research, and does not propose the research purpose of this paper based on the literature analysis.

You are right. I added some explanation of the research purpose in the manuscript.

Generally, elements that are analyzed in this research seem to be reasonable as an additives in the laser alloying to increase such properties of the surface layer as hardness, wear, corrosion or even heat resistance. As was found, the laser alloying treatment with those elements are subject of many studies. Those studies are explaining the influence of the particular additive on the surface layer under the specific treatment conditions.

Hence, it is difficult to evaluate the differences in effects between particular additives during laser alloying in the surface layer.

Therefore, by performing the same laser treatment conditions it is possible to compare the effects of different alloying substances on the surface layer, which was the purpose of this research.

  1. page3: Figure 3 should be Figure 2. How to ensure the reliability of the calculated temperature?

Number of figure is corrected.

(please note that the numbering of the figures has changed due to the addition of some)

The temperature was calculated on the base of equations presented by Ashby and Easterling [1]. These equations are used in a wide variety of laser processing studies as a help in assessing the potential effects of laser processing in the surface layer and in designing laser processing conditions. Obviously, the calculated temperature is only a simulation and the best assessment is a measurement. As shown in Figure 2, the maximum (calculated) temperature on the surface is approximately 2500ºC. This temperature level was indicated by the measurement during laser treatment with a pyrometer (Figure 4).

  1. The experimental details are not described in detail, such as sampling location, sample preparation method, etc.

Information on experimental details has been extended in the manuscript. Scheme of the sample has been added.

  1. Page4: “the heating can be divided into stages. This is due to the TEM01 mode of the laser device.” Please explain TEM01.

The mode expresses the laser beam power distribution.  There are many types of mode.

a)

You can find the figure in the Word document.

b)

You can find the figure in the Word document.

Figure: Types of transverse modes of the laser beam with axial symmetry (a) with examples of power distributions in the focus of the laser beam (b)

TEM01 means that when scanning with a laser beam, the maximum value of the power of the laser beam "passes" twice through each point in the center of the "laser path".

I added some explanation in the manuscript.

  1. Figure 5: How many samples of each condition were observed to guarantee the persuasiveness and reliability of the dimensions of the melted zone and the alloyed zones?

It was 4.

I added this information in the manuscript.

  1. What are the abscissa and ordinate of Figure 6a?

They are: intensity and energy. I added this to the figure.

  1. The image quality is too low, such as Fig8b and Fig 11.

I made a new figures.

  1. The conclusion is poorly written. The content is too long and repeats the previous results. The authors are advised to summarize the results.

You are absolutely right. I corrected.

Round 2

Reviewer 1 Report

The author has improved the manuscript considerably. In my view a few small recommendations are still necessary.

Materials and Methods

    1. Although you have indicated the composition of the etching reagent in the captions of the figures, it is necessary to indicate it in the materials and methods section when referring to the microstructure (lines 124 and 125). At the same time, it is necessary to indicate the percentage of nitric acid in the attack. For example Nital 4%.

Conclusions

1. Conclusions have improved, but a greater degree of synthesis is needed.

 My suggestion is to focus the conclusions as follows:

-Microstructure: martensitic structures (I don't know if there is some residual austenite) and ferritic structure (alloyed with cobalt).

-Thicknesses attained.

-Hardnesses achieved.

-Mechanical properties (modulus of elasticity, elastic modulus…)

-Final conclusion: the one with the best properties is (...) due to (...).

Omits phases obtained by XRD. You show it in the results already, and it is not a conclusion of your research. Remember that the conclusions must show in a summarised way the most important results of your research.

Author Response

Dear Reviewer,

Thank you again.

Comments and Suggestions for Authors

The author has improved the manuscript considerably. In my view a few small recommendations are still necessary.

Materials and Methods

  1. Although you have indicated the composition of the etching reagent in the captions of the figures, it is necessary to indicate it in the materials and methods section when referring to the microstructure (lines 124 and 125). At the same time, it is necessary to indicate the percentage of nitric acid in the attack. For example Nital 4%.

Done

Conclusions

  1. Conclusions have improved, but a greater degree of synthesis is needed.

 My suggestion is to focus the conclusions as follows:

-Microstructure: martensitic structures (I don't know if there is some residual austenite) and ferritic structure (alloyed with cobalt).

-Thicknesses attained.

-Hardnesses achieved.

-Mechanical properties (modulus of elasticity, elastic modulus…)

-Final conclusion: the one with the best properties is (...) due to (...).

Omits phases obtained by XRD. You show it in the results already, and it is not a conclusion of your research. Remember that the conclusions must show in a summarised way the most important results of your research.

I have arranged the conclusions according your suggestions and shortened them as I could. I hope, that now is ok.

Reviewer 2 Report

The author responsed most of the comments. However, almost no changes have been made to the conclusion section, it is still too long. 

Author Response

Dear Reviewer,

Thank you again.

Comments and Suggestions for Authors

The author responsed most of the comments. However, almost no changes have been made to the conclusion section, it is still too long. 

I have shortened them as I could. I hope, that now is ok